# Effect of Sprouting on Biomolecular and Antioxidant Features of Common Buckwheat (*Fagopyrum esculentum*)

**DOI:** 10.3390/foods12102047

**Published:** 2023-05-18

**Authors:** Sara Margherita Borgonovi, Elena Chiarello, Federica Pasini, Gianfranco Picone, Silvia Marzocchi, Francesco Capozzi, Alessandra Bordoni, Alberto Barbiroli, Alessandra Marti, Stefania Iametti, Mattia Di Nunzio

**Affiliations:** 1Department of Food, Environmental and Nutritional Sciences (DeFENS), University of Milan, Via Celoria 2, 20133 Milan, Italy; sara.borgonovi@unimi.it (S.M.B.); alberto.barbiroli@unimi.it (A.B.); alessandra.marti@unimi.it (A.M.); stefania.iametti@unimi.it (S.I.); 2Department of Agricultural and Food Sciences (DISTAL), University of Bologna, Piazza Goidanich 60, 47521 Cesena, Italy; elena.chiarello2@unibo.it (E.C.); federica.pasini5@unibo.it (F.P.); gianfranco.picone@unibo.it (G.P.); silvia.marzocchi4@unibo.it (S.M.); francesco.capozzi@unibo.it (F.C.); alessandra.bordoni@unibo.it (A.B.); 3Interdepartmental Centre for Industrial Agri-Food Research (CIRI), University of Bologna, Piazza Goidanich 60, 47521 Cesena, Italy

**Keywords:** buckwheat, germination, starch, protein hydrolysis, lipids, anti-nutritional factors, antioxidants, metabolome

## Abstract

Buckwheat is a pseudo-cereal widely grown and consumed throughout the world. Buckwheat is recognized as a good source of nutrients and, in combination with other health-promoting components, is receiving increasing attention as a potential functional food. Despite the high nutritional value of buckwheat, a variety of anti-nutritional features makes it difficult to exploit its full potential. In this framework, sprouting (or germination) may represent a process capable of improving the macromolecular profile, including reducing anti-nutritional factors and/or synthesizing or releasing bioactives. This study addressed changes in the biomolecular profile and composition of buckwheat that was sprouted for 48 and 72 h. Sprouting increased the content of peptides and free-phenolic compounds and the antioxidant activity, caused a marked decline in the concentration of several anti-nutritional components, and affected the metabolomic profile with an overall improvement in the nutritional characteristics. These results further confirm sprouting as a process suitable for improving the compositional traits of cereals and pseudo-cereals, and are further steps towards the exploitation of sprouted buckwheat as a high-quality ingredient in innovative products of industrial interest.

## 1. Introduction

Buckwheat (*Fagopyrum esculentum*) is a short-season crop that has recently attracted much interest because of its environmental adaptability [1]. Buckwheat grows well in low-fertility or acidic soils, does not need fertilizers or biocides, and is sustainable for organic and environmentally friendly farming [2]. Buckwheat is also relatively tolerant to UV-B radiation and drought if compared to other crops, due to the high levels of stress-mitigating phenolics [3]. Thus, more intensive exploitation of buckwheat could favor agricultural diversification and minimize environmental degradation, improving food and nutritional security [4]. From a nutritional standpoint, buckwheat has an appreciable protein content (7–21%), and is rich in dietary fibers and bioactive compounds that contribute to its good antioxidant capacity and have multiple positive implications for the consumers’ health [5]. In addition, buckwheat does not contain gluten and is emerging as an alternative to rice and corn in gluten-free formulations [6].

Despite the good nutritional value of buckwheat, the presence of a broad array of anti-nutritional components, such as phytic acid and inhibitors of digestive proteases, has impaired its exploitation [7]. Among the processes that could overcome this limitation, sprouting (or germination) has been reported as an effective and low-cost process with positive effects on the nutritional profile [8]. During sprouting, many biochemical modifications occur and modulate product characteristics, such as bioactivity and flavor [9]. Sprouting reactivates seed metabolism, leading to the catabolism and degradation of macronutrients and anti-nutritional compounds and the biosynthesis of secondary metabolites with potential health benefits, improving the nutritional and health value of sprouted seeds [10]. Actually, most of the studies conducted in buckwheat investigated the Impact of sprouting, taking into consideration only one or few aspects at a time as phenolic profiles and antioxidants [11], starch and protein [12], or fatty acids [13]. In addition, the different experimental conditions adopted in terms of time, temperature, and humidity during sprouting do not allow to have a uniform and comprehensive view of the impact of this technology on changes at the biomolecular level and to draw definitive conclusions.

This study aimed to provide an overview of the changes in the macro- and micromolecular profile and of the properties and content of bioactive compounds in sprouted buckwheat. Sprouting was performed at laboratory-scale level and under controlled conditions for 48 and 72 h to evaluate if the different sprouting time changes the impact of this technology on changes at the biomolecular level. Sprouting-related modifications in the content of anti-nutritional factors, in endogenous α-amylase and protease activity, and the modulation of key digestive enzymes were also investigated. A metabolomic approach (nuclear magnetic resonance, NMR) aimed at a comprehensive view of sprouting-related biochemical events by identifying and quantifying the major metabolites.

## 2. Materials and Methods

### 2.1. Materials

Hexane, isopropanol, methanol, chloroform, and isooctane were obtained from VWR Internationa (Radnor, PA, USA). Unless otherwise specified, chemicals and solvents were from Sigma-Aldrich (St. Louis, MO, USA), and were of the highest available analytical grade. Seeds from dehulled common buckwheat (*Fagopyrum esculentum*) were provided by Molino Filippini s.r.l. (Teglio, Italy), consisting of a mix of different varieties from Eastern Europe harvested at full maturity, even though the plant is still green, in August/September 2020.

### 2.2. Sprouting Process

Seeds (2.5 kg) were sprouted in a lab-scale climate chamber (IPP110ecoplus, Memmert GmbH + Co. KG, Schwabach, Germany). Seeds were soaked in water (1:3, *w*/*w*) for 16 h at 27 °C and sprouted for 48 h (BW-48) and 72 h (BW-72) at 27 °C and 90% relative humidity. Optimal sprouting times were selected on the basis of previous studies that demonstrated that excessive germination determined a substantial loss of flour properties, dough quality, and bread-baking performance [14,15]. After sprouting, seeds were dried at 50 °C for 8 h (Self-Cooking Centerfi, Rational International AG, Heerbrugg, Switzerland). Unsprouted seeds were used as the control (BW-0). All samples were milled into powder (<0.5 mm) in a laboratory mill (IKA Universalmühle M20; IKA Laborteknic, Staufen, Germany). Dry weight (dw) was measured according to AACC 08-01 method and was 0.87 g dw/g powder, 0.90 g dw/g powder, and 0.92 g dw/g powder for BW-0, BW-48, and BW-72, respectively.

### 2.3. Aqueous Extraction

To avoid interferences due to the vehicle, buckwheat powder was subjected in duplicate to aqueous extraction by suspending 1 g of sample in 10 mL 0.05 M sodium phosphate buffer pH 7 containing 0.1 M NaCl. After 1 h of stirring at 25 °C, the suspension was centrifuged (2500× *g*, 30 min, 25 °C), and the supernatant was stored at −18 °C.

### 2.4. Total Starch, Damaged Starch, Fiber, and Glucose Content

Total starch, damaged starch, and fiber content were measured according to AACC 76-13.01, 76-31.01 and AOAC 992-16 methods. Glucose content was evaluated using the D-Glucose Assay Kit (R-Biopharm, Darmstadt, Germany) following the manufacturer’s instructions. Values obtained were normalized for g dw in buckwheat.

### 2.5. Endogenous α-Amylase Activity

Endogenous α-amylase activity was evaluated using the Cereal α-Amylase Assay Kit (Megazyme, Wicklow, Ireland) following the manufacturer’s instructions.

### 2.6. SDS-PAGE Analysis of Soluble Proteins

A 0.1 mL volume of aqueous buckwheat extracts was incubated in an equal volume of 0.125 M tris(hydroxymethyl)aminomethane (Tris)-HCl, pH 6.8, 50% glycerol, 1.7% sodium dodecyl sulfate, 0.01% bromophenol blue, in the presence/absence of 0.5% (*v*/*v*) 2-mercaptoethanol, and heated at 95 °C for 5 min. Electrophoretic runs were carried out in 12% polyacrylamide gels. Gels were stained with Coomassie Blue R250 and grayscale-imaged on a benchtop scanner. For further analysis, images of each lane were vertically divided into four portions, according to their relative molecular mass (M_r_), total (M_r_ ≤ 97.4 k), high (97.4 < M_r_ ≤ 45 k), medium (45 < M_r_ ≤ 21.5 k), and low (M_r_ < 21.5 k). Image Lab software (Bio-Rad Laboratories, Hercules, CA, USA) was then used for quantitative image analysis.

### 2.7. Soluble Proteins and Amino Acids/Small Peptides Content

Soluble protein and amino acids/small peptides content in aqueous buckwheat extract were determined spectrophotometrically by the Bradford dye-binding assay [16] and by the o-phthaldialdehyde (OPA) assay for free amino groups [17] using bovine serum albumin and L-isoleucine as the respective standards.

### 2.8. Endogenous Protease Activity

Endogenous protease activity in aqueous buckwheat extracts was measured using azocasein as a non-specific substrate, according to de Freitas et al. [18], with slight modifications. The reaction mixture contained 0.5 mL of 1% azocasein in 0.5 mM Tris-HCl (pH 5.0) and 0.5 mL of buckwheat extract. The reaction was performed at 37 °C and stopped after 120 min by adding 1 mL of 10% trichloroacetic acid. After centrifugation (10,000× *g*, 10 min at 25 °C), 0.08 mL of 5 N NaOH was added to 0.4 mL of the supernatant, and the absorbance was measured at 420 nm.

### 2.9. Lipid Content and Composition

Total lipids were extracted from 0.1 g of buckwheat powder [19]. After methylation [20], the content and profile of fatty acids as methyl esters (FAMEs) were determined by fast GC (GC-2030AF; Shimadzu, Kyoto, Japan), using a capillary column (30 mt, 0.2 μm film thickness) with a programmed temperature gradient (50–250 °C, 10 °C/min) [21]. Chromatographic peaks were identified based on retention time from FAME standard mixture (Sigma-Aldrich, St. Louis, MO, USA) and quantitated using Lab Solution software (Shimadzu, Kyoto, Japan). Peroxidizability and unsaturation indexes were calculated as previously reported [22,23].

### 2.10. Lipid Peroxidation

Lipid peroxidation was assessed by quantifying conjugated dienes (CD) according to Situnayake et al. [24] with slight modifications. A buckwheat sample (100 mg) was treated with 3.6 mL of hexane/isopropanol (3:2, *v*/*v*), followed by thorough mixing and by the addition of 2.4 mL of a 7% solution (*w*/*v*) of anhydrous sodium sulfate. After phase separation, the upper layer was collected and evaporated under nitrogen. The lipid residue was dissolved in 5 mL of isooctane, and its absorbance was read at 232 nm against an appropriate blank.

### 2.11. Phytic Acid Content

Phytic acid was estimated by the Phytic Acid/Total Phosphorus Kit (Megazyme International Ltd., Bray, Ireland) according to the manufacturer’s instructions.

### 2.12. Pepsin, Trypsin, and Chymotrypsin Activity

Pepsin, trypsin, and chymotrypsin activities were determined according to Urbinati et al. [25], with slight modifications. Bovine blood hemoglobin (0.48 mL, 2% solution at pH 2 for pepsin and pH 7 for trypsin and chymotrypsin) was added to 0.02 mL of aqueous buckwheat extract. The required enzyme was then added (0.1 mL, 0.03 mg/mL) to start the reaction, which was stopped after 10, 20, or 30 min by adding 1 mL of 20% (*w*/*v*) trichloroacetic acid. Soluble peptides in the supernatant after centrifugation (14,000× *g*, 10 min at 25 °C) were detected spectrophotometrically at 280 nm. Readings were corrected by endogenous proteolytic activity, measured without added digestive enzymes.

### 2.13. Tocols Extraction and Determination by HPLC–FLD

Tocols were determined as previously reported [26]. Then, 100 mg of extracted lipids were dissolved in 1 mL hexane and filtered through a 0.2 μm nylon filter. A 2.5 μL aliquot of the hexane solution was injected in a HPLC 1200 series equipped with a fluorimeter detector (λ_ex_ = 290 nm, λ_em_ = 325 nm) (Agilent Technologies, Palo Alto, CA, USA) fitted with a HILIC Poroshell 120 (3 × 100 mm, 2.7 μm) from Agilent Technologies (Palo Alto, CA, USA). An n-hexane/ethyl acetate/acetic acid (97.3:1.8:0.9 *v*/*v*/*v*) mobile phase was used for isocratic elution (0.8 mL/min). A calibration curve was constructed with α-tocopherol (Sigma-Aldrich, St. Louis, MO, USA), as previously reported [26,27,28,29].

### 2.14. Extraction and Determination of Free and Bound Phenolic Compounds

Free phenolics compounds were extracted twice from 2 g of buckwheat powder using ethanol/water (4:1, *v*/*v*) in an ultrasonic bath [30]. The supernatants were collected, evaporated, and reconstituted with 2 mL of methanol/water (1:1, *v*/*v*). The extracts were stored at −18 °C until use. Residues of free phenolic extraction were shaken overnight (20 h) with 200 mL of 2 M NaOH at room temperature under nitrogen to obtain the bound phenolic fraction. The hydrolyzed solution was acidified to pH 2 by adding 10 M hydrochloric acid in an ice bath. The final solution was extracted five times with 100 mL of ethyl acetate, and the pooled organic fractions were evaporated to dryness. The bound phenolic compounds were reconstituted in 2 mL of methanol/water (1:1, *v*/*v*). Separation of free and bound phenolic compounds from buckwheat powder was carried out using a C-18 column (Poroshell 120, SB-C18, 3.0 × 100 mm, 2.7 µm from Agilent Technologies, Palo Alto, CA, USA) and an Agilent HPLC 1200 series equipped with auto-sampler and a binary pump, according to the methods reported by Gomez-Caravaca et al. [31]. MS/MS analysis (MRM mode) was performed on 6420 Triple Quadrupole (Agilent Technologies, Santa Clara, CA, USA) using an electrospray ionization (ESI) interface in negative and positive ionization mode. Ferulic acid, catechin, and rutin were used as standards for quantitative purposes.

### 2.15. Total Antioxidant Capacity (TAC) and Ferric Reducing Antioxidant Power (FRAP)

TAC and FRAP were assessed by measuring the ability of the antioxidant molecules in the sample to reduce the radical cation of 2,2′-azinobis-(3-ethylbenzothiazoline-6-sulfonic acid) (ABTS^•+^) [32] and the Fe (III)/tripyridyltriazine complex [33], respectively, using 0.01 mL of the aqueous buckwheat extract. Values obtained were compared to the concentration–response curve of a standard solution of 6-hydroxy-2,5,7,8-tetramethylchroman-2-carboxylic acid (Trolox).

### 2.16. ^1^H(HR)-NMR Spectra Acquisition and Processing

Aqueous buckwheat extracts were thawed and centrifuged at 2300× *g* for 5 min at 4 °C to eliminate the coarsest particles and subsequently at 50,000× *g* for 5 min at 4 °C to eliminate the finest particles. A 0.75 mL aliquot of the supernatant was added to 0.12 mL of 100 mM phosphate buffer containing 10 mM trimethylsilylpropanoic acid (TSP, as internal standard) and brought to pH 7. HR-NMR spectra were recorded at 298 K on a Bruker US+ Avance III spectrometer, as reported elsewhere [26]. Signals were identified by comparing their chemical shift and multiplicity with Chenomx Profiler software data bank (ver. 8.1, Edmonton, AB, Canada). Before statistical analysis, the NMR spectra underwent several pre-processing procedures, such as spectra alignment, removal of some irrelevant signals, normalization, and a final binning [34]. Some parts of spectra lacking metabolic information were removed, including (i) the noise-only regions from 20.00 to 9.40 ppm and from −20.00 to −0.50 ppm; and (ii) the region from 4.69 to 5.05 ppm where water was highly interfering. The new dataset was normalized by applying the Probabilistic Quotient Normalization (PQN) [35], based on the calculation of a most probable dilution factor by looking at the distribution of the quotients of the amplitudes of the samples’ spectra to reference one. Further crucial data reduction was performed by using a binning (or bucketing) algorithm [36]. Spectra were reduced to 348 bins of 150 data points, each bin corresponding to a spectral region of 0.0274 ppm.

### 2.17. Statistical Analysis

Statistically significant differences were determined by one-way analysis of variance (ANOVA) followed by Tukey’s post hoc test and considering *p* < 0.05 as significant. After pre-processing, NMR data underwent multivariate analyses (PCA) first and then univariate (ANOVA and Tukey’s post hoc test). Statistical analyses were carried out by using the R software environment for statistical computing (version 4.1.0).

## 3. Results and Discussion

### 3.1. Carbohydrates

As reported for cereals, pseudo-cereals, and legumes [37,38,39], sprouting significantly promoted the hydrolysis of starch to glucose due to α-amylase activation in the scutellum and aleurone in response to providing energy for seed development [40]. Additionally, in the case of buckwheat, the total starch content decreased with increasing sprouting time (Figure 1A), leading to a release of free glucose (Figure 1B). The increase in damaged starch confirms the starch hydrolysis (Figure 1C), which is an indicator of susceptibility to α-amylase hydrolysis [41], and endogenous α-amylase activity as well (Figure 1D).

Furthermore, damaged starch, which represents the fraction of starch readily accessible to amylase hydrolysis, increased in sprouted buckwheat, likely due to the presence of some holes in the outermost regions of the granules, as observed in other species [42]. Together, these results could suggest an increased starch digestibility in sprouted flours. In support of this hypothesis, Sharma et al. [43] recently showed in millet that sprouting gradually lowers the fractions of resistant and slowly digestible starch while increasing the rapidly digestible starch. More recently, Molska et al. [12] demonstrated that sprouted buckwheat possesses a higher rate of starch hydrolysis during in vitro digestion. Cooking is needed before consumption of starch-based foods and this process, together with milling, modifies the digestibility of the starch [44,45]. Therefore, more studies evaluating the digestibility of sprouted grain foods are needed before conclusions can be drawn. Anyway, it is noteworthy that despite the increased glucose content, in vitro and in vivo studies have reported that sprouted grains do not alter the glycemic index and improve fasting blood glucose, making them a good candidate for blood sugar control [46,47]. Sprouting within 72 h did not modify the content of dietary fibers. This confirms that starch is the main glucidic fraction targeted by endogenous hydrolytic enzymes at the sprouting time considered in buckwheat.

### 3.2. Proteins

In buckwheat proteins, the percentage of albumins and globulins (45%) is higher than in cereal proteins, with a matching lower content in glutelins (15%) and prolamins (3%) [48]. Buckwheat globulins consist of a major 13S legumin-like and a minor 8S vicilin-like fractions made up of proteins with M_r_ ranging from 68 kDa to 26 kDa. Buckwheat albumins represent about 25% of the total proteins and consist mainly of single-chain polypeptides with M_r_ in the 8–16 kDa range [49,50].

The various panels of Figure 2 report the SDS-PAGE tracings of soluble proteins present in buckwheat aqueous extracts when run in the absence (Figure 2A) and in the presence (Figure 2C) of disulfide reducing agents. Confirming previous reports, the most substantial protein bands in unsprouted buckwheat had at M_r_ values of 67, 35, 21, and 16 kDa under non-reducing conditions [51]. Under reducing conditions, the bands at M_r_ 67 and 35 kDa disappeared and were replaced by novel bands at M_r_ around 55, 25, and 10 kDa.

Sprouting was accompanied by a time-dependent proteolysis of the large soluble aggregates evident under non-reducing conditions to produce species of M_r_ around 40 kDa, likely along with smaller peptides that may have escaped detection. The 40 kDa species were not present in the extracts from unsprouted buckwheat and were not observed in the presence of disulfide reductants, suggesting they are formed by disulfide bound peptides originating from nicking of the original aggregates upon sprouting.

As shown in Figure 2B, the largest peptides evident in the unsprouted sample under reducing conditions were quite insensitive to sprouting-dependent proteolysis, and there was a marked difference with the time-dependent decrease in the intensity of smaller bands. The progressive disappearance of the polypeptide at M_r_ around 35 kDa is particularly striking, as this component appears to be present in a nicked form at 48 h and to be almost completely degraded at 72 h. Finally, it is remarkable that the endogenous proteases activated during sprouting can break down even the smallest proteins in buckwheat, regardless of whether they were present in a free-living form rather than disulfide-linked to larger proteins. Indeed, analysis of band intensity in the regions that comprise polypeptides with M_r_ < 21.5 kDa (Figure 2B,D) indicates that the products of the proteolytic breakdown of larger species did not accumulate. On the contrary, they were progressively disappearing—at the equivalent rate—in samples analyzed in the presence and absence of disulfide reductants.

The overall pattern of events hypothesized above was confirmed by dye-binding and OPA assays, which can detect large proteins and small peptides, respectively. Results of the two assays at various sprouting times evidenced how a time-dependent decrease in the content of soluble proteins (M_r_ > 3 kDa, Figure 3A) [52] was accompanied by increased release of very small peptides and/or individual amino acids [53] (Figure 3B). The measurement of the endogenous protease activity on a convenient non-plant-protein substrate (Figure 3C) confirmed that the activity of endogenous proteases increased steadily in seeds during the sprouting period considered in this study. In buckwheat, protease activation was reported to increase up to day four of germination [54]. As in other grains, protein breakdown is essential to provide the amino acids required for embryo growth and plant development [8]. The nature, number, specificity, and activation mechanism of endogenous enzymes involved in this process remains to be assessed in buckwheat and other grains.

### 3.3. Lipids

As previously reported [55], the main fatty acids in buckwheat were linoleic > oleic > palmitic acid, which, together, accounted for approximately 90% of total fatty acids. Sprouting modulated the fatty acid composition of the buckwheat powder (Table 1). The sprouting-related decrease in total fatty acid content targeted mostly saturated fatty acids (SFA) and monounsaturated fatty acids (MUFA), and it was likely related to the re-activation of β-oxidation in the glyoxysomes to fulfill the energy needs of the growing seed [56]. It is of note that germination did not modify the total content of polyunsaturated fatty acids (PUFA), presumably due to their preferential use as a structural component of cellular membranes [57], thus imparting a healthier profile to the fats in sprouted buckwheat although increasing their peroxidability. The unsaturation index (UI) and peroxidability index (PI) increased in sprouted buckwheat (Table 1). A representative chromatogram of BW-0 has been included as Appendix A.

Lipid peroxidation was evaluated by monitoring the concentration of CD-containing lipids [58], which increased significantly during sprouting. It is conceivable that the increased lipid peroxidation reflects the transition from seed dormancy to germination, a physiological process regulated by diverse endogenous factors, including reactive oxygen species, which promote the release of seed dormancy by biomolecules oxidation, testa weakening, and endosperm decay [59].

### 3.4. Bioactive Compounds

#### 3.4.1. Tocols

Although buckwheat exhibits levels of tocopherols similar to wheat, barley, oats, and rye, with γ-tocopherol being the main isoform present, typically in a >10-fold excess with respect to α- and δ-tocopherols [60], buckwheat and corn bran, and wheat germ were dominated by tocopherols, whereas the oat, rice, rye, spelt, and wheat bran oils were rich in tocotrienols [61]. Accordingly, in this study, the main tocols in buckwheat were γ- > δ- ≃ α-tocopherol (Table 2). Sprouting did not affect the α-, γ-, and total tocopherol levels but showed a slight but significant decrease in the δ -isoform. Previous studies also reported sprouting-related changes in the content of tocopherols homologues [62] in various cereals [63,64].

#### 3.4.2. Free and Bound Polyphenols

Phenolic compounds are present in plants either in free or bound form, the latter most commonly being ester-linked to structural cell wall polymers [65]. Although the major portion of phenolics in grains is in the bound form [66], buckwheat contains most of its phenolic compounds in the more bio-accessible free form [67,68].

In this study, 29 free phenolic compounds were identified and quantified in buckwheat. As summarized in Table 3, the buckwheat phenolics were representative of five classes, 6 phenolic acids; 15 flavan-3-ols; 3 flavonols; 3 flavones; and 2 proanthocyanidins. Among them, six were also found in a bound form (four phenolic acids, one flavonol, and one flavone). In unsprouted buckwheat, the most representative free phenolic classes were flavan-3-ols > proanthocyanidins > flavones, all together accounting for about 96% of the total free phenolics content. The main classes found in the bound form were phenolic acids and flavones, which accounted for about 92% of total bound phenolics. The total content in free phenolic compounds was 30 times higher than that of the bound species. The resulting relative abundance was in the order flavan-3-ols > free proanthocyanidins > free flavones > free phenolic acids > bound phenolic acids > free flavonols > bound flavones > bound flavonols > bound flavan-3-ols = bound proanthocyanidins.

In 48 h and 72 h sprouted buckwheat, the primary free phenolic classes were flavones > flavan-3-ols > proanthocyanidins, accounting for approximately 98% of total free phenolic compounds. The main bound phenolic classes were flavones, which accounted for about 97% of total bound phenolic compounds. Total free phenolic compounds were approximately 11 and 8 times higher than the bound counterpart in 48 and 72 h sprouted buckwheat, respectively. The main phenolic classes in 48 h sprouted buckwheat were free flavones > free flavan-3-ols > bound flavones > free proanthocyanidins > free phenolic acids > free flavonols > bound phenolic acids > bound flavan-3-ols = bound flavonols = bound proanthocyanidins. The main phenolic classes in 72 h sprouted buckwheat were free flavones > bound flavones > free flavan-3-ols > free proanthocyanidins > free phenolic acids > free flavonols > bound phenolic acids > bound flavan-3-ols = bound flavonols = bound proanthocyanidins. Sprouting determined an increase in total free phenolic acids, total free flavonols, total free flavones, total free proanthocyanidins, total free phenols compounds, total bound flavones, and total bound phenols compounds content. On the contrary, sprouting caused a diminished content of total free flavan-3-ol, total bound phenolic acids, and total bound flavonols.

De novo synthesis and transformation could be responsible for the dramatically increased levels of polyphenols during germination. The primary building block for the synthesis of phenolic compounds is glucose, and several crucial molecular signaling pathways, including the oxidative pentose phosphate pathway, glycolysis, acetate/malonate pathway, shikimate pathway, phenylpropanoid pathway, and hydrolyzable tannin pathway, are involved in the synthesis and transformation of polyphenols during the earliest phases of plant growth [69]. Although we have considered the seeds as a whole, a previous study showed that polyphenols compounds are mainly accumulated in cotyledons and hypocotyls during buckwheat germination [70]. In addition, de novo synthesis could result from the activation of phenylalanine ammonia lyase (PAL), the key enzyme in phenolic biosynthesis involved in forming phenylpropanoids, hydroxycinnamates, flavonoids, proanthocyanidins, hydroxystilbenes, coumarins, lignans, and lignins [71]. During buckwheat germination, a positive linear correlation between PAL activity and flavonoids and phenolic accumulation was evidenced, suggesting that the variation in PAL activity was probably involved with phenolic (or flavonoid) accumulation [72].

### 3.5. Antioxidant Capacity

The sprouting-dependent increase in the content of phenolics also brought forward an increase in antioxidant capacity. As shown in Table 4, FRAP in aqueous buckwheat extracts increased progressively during sprouting, whereas TAC was statistically significant only after 72 h. Albeit not statistically significant, a close linear correlation was present between TAC and free phenolic content (Pearson r = 0.996, r^2^ = 0.992, *p* = 0.057). The feebler correlation between free phenolic content and antioxidant activity is probably because the antioxidant activity results from a different type of extractable bioactive component with antioxidant activity, such as citric, ferulic, and ascorbic acids, frequently present in cereal seed [73]. In addition to the polyphenols content, the structure–activity relationship and the in vitro assay adopted should be also deeply considered in evaluating the polyphenols’ antioxidant activity [74]. Hydroxyl groups on ring-B and the presence of a 3-hydroxyl group on ring-C in flavonoids increased TAC and FRAP but phenolic acids lacking a 3-hydroxyl group had significantly lower FRAP [75].

Augmented levels in antioxidants, mainly in the most bioaccessibility-free form of polyphenols, may have a major nutritional backlash related to their antioxidant ability. Merendino et al. [76] evidenced in spontaneously hypertensive (SHR) and Wistar-Kyoto rats fed with pasta made with 30% of sprouted buckwheat powder an amended plasma antioxidant capacity and reduced oxidative markers and genotoxic effects respect rats fed with commercial pasta. Moreover, oxidatively stressed SHR rats fed with sprouted buckwheat powder-enriched pasta also determined a significant decrease in DNA damage and a more efficient DNA repair than the control diet [77].

### 3.6. Metabolome

The NMR spectroscopy is an essential tool that provides information for the molecular characterization of natural products due to its intrinsic ability of quantifying all detectable components in complex mixtures, directly without a preliminary separation. Particularly, ^1^H NMR-based metabolomics have proven effective and efficient because ^1^H atoms are ubiquitous and in high isotopic abundance, thus allowing high-throughput acquisition of spectra to identify and quantify most metabolites [78]. Using an untargeted approach, it was tested whether sprouting determined a time-dependent increase in free amino acid, sugars, and organic acid levels, indicating the re-activation of seed metabolism upon germination.

The effect of sprouting on the metabolome has been evaluated by an NMR-based metabolomics approach in combination with Foodomics and Chemometrics techniques [79]. Unsupervised principal component analysis was carried out on a binned dataset. The obtained model was used to select metabolites mainly involved in the source of diversification between samples collected in the whole experimental set, by applying the ANOVA on the total bins. This approach allowed for assessing the time course of changes in the metabolic profile at various sprouting times. A total of 15 metabolites, including 7 free amino acids, 3 sugar, 3 organic acids, 1 nucleotide, and 1 nicotinic acid derivative were identified throughout the ^1^H NMR spectra. Table 5 lists the essential bins that changed their integral area as the concentration levels of the corresponding metabolites changed during sprouting. In summary, sprouting is associated with an increased concentration of tryptophan, glutamine, alanine, valine, isoleucine, glucose, fructose, acetate, and lactate, whereas a decrease in trigonelline concentration was observed upon sprouting.

Although NMR has just been used to identify the major metabolites in sprouted legumes [80,81], to the best of our knowledge, this is the only study that used it to evaluate the impact of sprouting on the metabolome in germinated buckwheat. The increased glucose levels in the sprouted samples confirm the relevance of starch hydrolysis during germination. In addition, the increased levels of fructose, acetate, and lactate appear likely associated with a burst in glucose metabolism and enzymes activation [54,82]. Along the same line of reasoning, substantial endogenous proteolysis during sprouting increases (almost 3-fold, at 72 h) the levels of several free amino acids and glutamine, the latter is an amino acid abundant in storage proteins and relevant to nitrogen metabolism. These results indicate that metabolites dynamics during seeds germination are pretty complicated as substances are continuously synthesized, recycled, and degraded due to metabolism reactivation. This demonstrates that analyzing the chemical diversity and the wide range of metabolite concentrations in plants necessitates authoritative analytical approaches. In this complicated but intriguing scenario, ^1^H-NMR spectroscopy could be considered a powerful analytical tool, offering the opportunity for reliable metabolite detection and investigating the whole set of metabolites, which are essentially nongenetically encoded substrates, intermediates, and products of biochemical pathways [81,83,84].

### 3.7. Anti-Nutritional Factors

Sprouting decreased steadily the content of phytic acid but the analytical difference was statistically significant only after 72 h of germination (Table 6).

The process of sprouting can decrease phytate concentrations present through the activation and de novo synthesis of phytase, which release myo-inositol, phosphate, and other minerals for plant growth [85], as well as the leaching of water-soluble phytate during soaking [86].

Since cereal grains are a natural source of protease inhibitors [87], the effect of sprouting on the activity of the major digestive enzymes was evaluated to the same enzymes without buckwheat extracts. As shown in Table 7, unsprouted buckwheat aqueous extracts almost blocked trypsin activity, and the removal of trypsin inhibitors upon sprouting was non-linear, with more than 50% inhibitory capacity still present at 72 h of sprouting. Chymotrypsin activity was not affected by the addition of any of the buckwheat extracts, whereas BW-72 extract significantly enhanced pepsin activity.

Although various studies evidenced an ameliorative effect of sprouting on trypsin inhibition and phytates content due to enzymatic degradation [88,89,90], to the best of our knowledge, this is the first study evaluating the effect of sprouting on pepsin and chymotrypsin activity in buckwheat. The impact on digestive proteases reported here confirms recent reports on improved in vitro protein digestibility in sprouted brown finger millet [91] and sprouted wheat [92]. These results appear to be of particular importance in light of the use and exploitation of germinated cereals to obtain foods with a higher digestibility.

## 4. Conclusions

Sprouting caused significant changes in the composition of buckwheat seeds. While some changes were already maximal after 48 h, 72 h appeared to be the best sprouting duration after this method had been adopted. The increased release of free peptides/amino acids and phenolic compounds with antioxidant activity, and the substantial decrease in anti-nutritional factors, suggest sprouting as a suitable process to improve buckwheat nutritional properties, thus obtaining a high-quality ingredient. With this in mind, the final food products should carefully monitor some changes observed after sprouting. In fact, increased perishability could result from increased oxidation of lipids.

Further studies, including evaluating the digestibility of sprouted material, the presence of bioactive species in proteolytic fragments, and the effect on the gut microbiota are needed to boost the industrial exploitation of buckwheat and traditional buckwheat foods.

## Figures and Tables

**Figure 1 foods-12-02047-f001:**
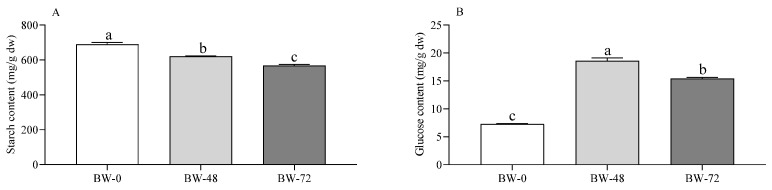
Starch content (**A**), glucose content (**B**), damaged starch content (**C**), and endogenous α-amylase activity (**D**) in unsprouted buckwheat (BW-0), and sprouted buckwheat (48 h, BW-48; 72 h, BW-72). Data are expressed as mg/g dw (**A**–**C**) and U/g dw (**D**) and are mean ± SD of two different extractions and at least duplicate assays on each extract. Statistical analysis was by one-way ANOVA (always: *p* < 0.05) with Tukey’s post hoc test (different letters in the same panel indicate significant differences).

**Figure 2 foods-12-02047-f002:**
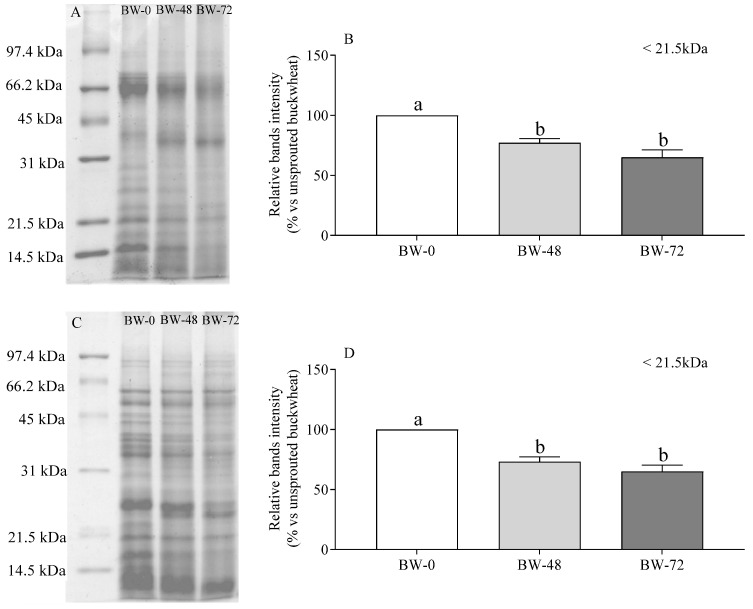
SDS-PAGE tracings (**A**,**C**) and relative band intensity at M_r_ < 21.5 kDa (**B**,**D**) of proteins in the absence (**A**,**B**) and in the presence (**C**,**D**) of disulfide reducing agents in aqueous extracts from unsprouted (BW-0), and sprouted buckwheat (48 h, BW-48; 72 h, BW-72). Band intensity is expressed as the percent of unsprouted buckwheat (assigned as 100%), considering at least two individual SDS-PAGE runs for each condition. Statistical analysis was by one-way ANOVA (always: *p* < 0.05) with Tukey’s post hoc test (different letters indicate significant differences).

**Figure 3 foods-12-02047-f003:**
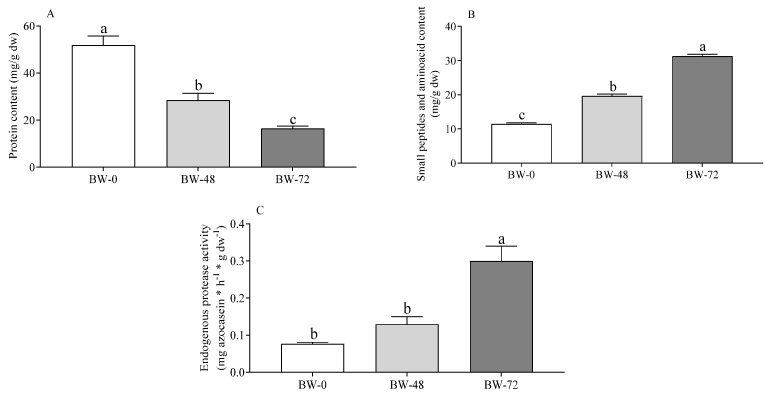
Soluble protein (**A**), peptides/amino acids (**B**), and endogenous protease activity (**C**) in unsprouted (BW-0) and sprouted buckwheat (48 h, BW-48; 72 h, BW-72). Data are expressed as mg/g dw (**A**,**B**), and mg azocasein/h/g dw (**C**) and are means ± SD of two different extractions and at least duplicate assays. Statistical analysis was by one-way ANOVA (always: *p* < 0.05) with Tukey’s post hoc test (different letters indicate significant differences).

**Table 1 foods-12-02047-t001:** Fatty acid methyl esters and conjugated diene content in unsprouted (BW-0) and sprouted buckwheat (48 h, BW-48; 72 h, BW-72).

FAME	BW-0	BW-48	BW-72
C14:0	0.07 ± 0.01a	0.05 ± 0.00a	0.03 ± 0.04a
C16:0	3.96 ± 0.02a	3.30 ± 0.18b	2.81 ± 0.34c
C16:1 n-7	0.07 ± 0.10a	0.05 ± 0.00a	0.04 ± 0.06a
C17:0	0.03 ± 0.04a	0.04 ± 0.00a	0.02 ± 0.03a
C18:0	0.21 ± 0.07a	0.08 ± 0.03b	0.04 ± 0.05c
C18:1 n-9	8.09 ± 0.39a	6.59 ± 0.36a	4.72 ± 0.97b
C18:2 n-6	8.79 ± 0.60a	9.14 ± 0.59a	7.63 ± 1.25a
C18:3 n-3	0.52 ± 0.02b	0.68 ± 0.04a	0.65 ± 0.04a
C20:0	0.32 ± 0.06a	0.28 ± 0.01a	0.22 ± 0.09a
C20:1 n-9	0.68 ± 0.09a	0.59 ± 0.01a	0.39 ± 0.07b
C22:0	0.46 ± 0.19a	0.31 ± 0.01a	0.27 ± 0.11a
ΣSFA	5.04 ± 0.27a	4.06 ± 0.17b	3.39 ± 0.42b
ΣMUFA	8.84 ± 0.38a	7.23 ± 0.37a	5.15 ± 0.98b
ΣPUFA	9.31 ± 0.62a	9.82 ± 0.63a	8.28 ± 1.3a
Σn-6/Σn-3	16.74 ± 0.49a	13.46 ± 0.12b	11.65 ± 1.15b
Total	23.20 ± 1.27a	21.11 ± 1.17ab	16.81 ± 2.70b
UI	120.65 ± 0.50c	130.47 ± 0.65b	132.97 ± 0.16a
PI	43.35 ± 0.44c	50.57 ± 0.39b	53.92 ± 0.55a
CD	100.00 ± 2.65c	174.03 ± 20.29b	252.00 ± 20.40a

Fatty acids and conjugated dienes content are expressed as mg FAME/g dw and as percent (%) of unsprouted buckwheat flour (assigned as 100%), respectively. Data are means ± SD of two different extractions and duplicate gas-chromatographic analysis. Statistical analysis was by one-way ANOVA (C16:0, C18:0, C18:1 n-9, C18:3 n-3, C20:1 n-9, ΣSFA, ΣMUFA, Σn-6/Σn-3, total FAME, PI, UI, CD: *p* < 0.05) with Tukey’s post hoc test (different letters indicate significant differences). CD: conjugated diene; FAME: fatty acid methyl esters; SFA: saturated fatty acids; MUFA: monounsaturated fatty acids; PUFA: polyunsaturated fatty acids; PI: peroxidability index; UI: unsaturation index.

**Table 2 foods-12-02047-t002:** Tocols content in unsprouted (BW-0) and sprouted buckwheat (48 h, BW-48; 72 h, BW-72).

Tocols ^1^	BW-0	BW-48	BW-72
α-tocopherol	2.11 ± 0.10a	2.70 ± 0.32a	2.48 ± 0.12a
γ-tocopherol	56.07 ± 0.16a	55.31 ± 5.03a	44.55 ± 1.61a
δ-tocopherol	3.79 ± 0.03a	3.52 ± 0.05b	2.95 ± 0.03c
Total tocols	61.97 ± 0.26a	61.53 ± 5.2a	49.98 ± 2.10a

Data are expressed as µg/g dw and are means ± SD of two extractions and duplicate chromatographic runs. Statistical analysis was by one-way ANOVA (δ-tocopherol: *p* < 0.05) with Tukey’s post hoc test (different letters indicate significant differences). ^1^ Calculated using the calibration curve of α-tocopherol.

**Table 3 foods-12-02047-t003:** Phenols content in unsprouted (BW-0) and sprouted buckwheat (48 h, BW-48; 72 h, BW-72).

Compounds	[M − H]^−^	MS Fragments	Q. T.	Free Phenolic Compounds			Anova	Bound Phenolic Compounds			Anova
				BW-0	BW-48	BW-72		BW-0	BW-48	BW-72	
Phenolic acids											
Protocatechuic-4-*O*-glucoside acid	315	153	315→153	30.25 ± 1.01c	59.31 ± 0.73b	82.90 ± 4.24a	*p* < 0.05	11.92 ± 0.09a	3.39 ± 0.61b	2.93 ± 0.10b	*p* < 0.05
Caffeic acid hexose	341	251	341→251	1.88 ± 0.22a	2.15 ± 0.03a	2.09 ± 0.25a	n.s.	n.d.	n.d.	n.d.	
Caffeic acid hexose	341	251	341→251	n.d.c	5.82 ± 0.85b	13.30 ± 0.56a	*p* < 0.05	n.d.	n.d.	n.d.	
*p*-Coumaric acid	163	119	163→119	3.25 ± 0.60b	8.76 ± 0.52a	9.08 ± 0.92a	*p* < 0.05	n.d.b	0.25 ± 0.04b	3.28 ± 0.59a	*p* < 0.05
Swertiamacroside	487	451, 179	487→179	0.93 ± 0.15b	3.95 ± 0.67a	4.44 ± 0.62a	*p* < 0.05	22.99 ± 1.64a	10.37 ± 0.67b	15.48 ± 1.22b	*p* < 0.05
Ferulic acid	193	178	193→178	n.d.c	2.49 ± 0.15b	3.84 ± 0.49a	*p* < 0.05	n.d.b	0.32 ± 0.03b	3.67 ± 0.37a	*p* < 0.05
Total phenolic acids				36.31 ± 1.53c	82.48 ± 1.43b	115.65 ± 2.82a	*p* < 0.05	34.90 ± 1.55a	14.34 ± 1.27c	25.36 ± 1.54b	*p* < 0.05
Flavan-3-ols											
Catechin-glucoside	451	289	451→289	118.76 ± 2.96c	202.06 ± 28.76b	356.54 ± 0.70a	*p* < 0.05	n.d.	n.d.	n.d.	
Catechin	289	203	289→203	0.87 ± 0.05c	36.42 ± 2.88b	60.88 ± 5.11a	*p* < 0.05	n.d.	n.d.	n.d.	
(Epi)afzelchin-(epi)catechin isomer A	561	543, 435, 425, 407, 289, 271	561→289	38.70 ± 1.04a	2.12 ± 0.06c	6.78 ± 0.23b	*p* < 0.05	n.d.	n.d.	n.d.	
Catechin-glucoside	451	289	451→289	95.00 ± 0.88b	117.91 ± 2.69a	35.68 ± 3.91c	*p* < 0.05	n.d.	n.d.	n.d.	
Epicatechin	289	244	289→244	59.42 ± 1.13a	20.65 ± 0.21b	16.25 ± 1.36c	*p* < 0.05	n.d.	n.d.	n.d.	
Catechin-glucoside	451	289	451→289	17.66 ± 2.96b	38.20 ± 6.74a	29.02 ± 2.10ab	*p* < 0.05	n.d.	n.d.	n.d.	
(Epi)afzelchin-(epi)catechin isomer B	561	543, 435, 425, 407, 289, 271	561→289	124.29 ± 0.46a	55.97 ± 1.6b	21.66 ± 1.41c	*p* < 0.05	n.d.	n.d.	n.d.	
Epiafzelchin-epiafzelchin-epicatechin	833	561, 543, 289, 271	833→561	143.23 ± 9.48a	140.22 ± 0.02a	125.91 ± 2.01a	n.s	n.d.	n.d.	n.d.	
(Epi)afzelchin-(epi)catechin isomer C	561	543, 435, 425, 407, 289, 271	561→289	45.04 ± 8.17a	10.89 ± 2.09b	13.65 ± 0.63b	*p* < 0.05	n.d.	n.d.	n.d.	
Epicatechin-gallate	441	289, 169	441→169	9.62 ± 1.12b	20.15 ± 1.38a	10.18 ± 0.04b	*p* < 0.05	n.d.	n.d.	n.d.	
Epiafzelchin-epicatechin-*O*-methyl gallate	727	561, 455, 289, 271	727→289	125.94 ± 3.21a	128.91 ± 1.63a	119.97 ± 5.65a	n.s.	n.d.	n.d.	n.d.	
(-)-Epicatechin-3-(3″-*O*-methyl) gallate	455	289, 183	4555→183	149.20 ± 1.76a	48.49 ± 7.36b	43.41 ± 0.79b	*p* < 0.05	n.d.	n.d.	n.d.	
(Epi)afzelchin-(epi)catechin isomer D	561	543, 435, 425, 407, 289, 271	561→289	32.32 ± 2.90a	8.32 ± 0.23b	0.67 ± 0.03c	*p* < 0.05	n.d.	n.d.	n.d.	
Epiafzelchin-epicatechin-*O*-dimethyl gallate	741	469, 319, 271	741→469	78.74 ± 4.91a	32.42 ± 0.97b	34.20 ± 1.14b	*p* < 0.05	n.d.	n.d.	n.d.	
Epicatechin-*O*-3,4-dimethyl gallate	469	319, 271, 125	469→271	126.39 ± 4.50a	25.58 ± 0.79b	19.49 ± 1.08b	*p* < 0.05	n.d.	n.d.	n.d.	
Total flavan-3-ol				1165.17 ± 14.29a	888.32 ± 37.45b	894.29 ± 13.16b	*p* < 0.05	n.d.	n.d.	n.d.	
Flavonols											
Quercitrin	447	301, 179, 151	447→301	4.64 ± 0.50b	3.20 ± 0.13b	8.61 ± 1.41a	*p* < 0.05	n.d.	n.d.	n.d.	
Rutin	609	301	609→301	9.91 ± 0.84c	15.59 ± 0.14b	23.01 ± 1.15a	*p* < 0.05	n.d.	n.d.	n.d.	
Quercetin	301	178, 151	30→151	3.39 ± 0.15b	21.37 ± 1.02a	28.12 ± 2.93a	*p* < 0.05	4.29 ± 0.01a	n.d.b	n.d.b	*p* < 0.05
Total flavonols				17.93 ± 1.19c	40.17 ± 1.29b	59.74 ± 2.66a	*p* < 0.05	4.29 ± 0.01a	n.d.b	n.d.b	*p* < 0.05
Flavones											
Orientin	447	357, 327	447→357	2.42 ± 0.18c	16.27 ± 0.54b	24.20 ± 1.52a	*p* < 0.05	n.d.	n.d.	n.d.	
Isorientin	447	357, 327	447→357	9.56 ± 1.73c	403.42 ± 22.51b	1419.45 ± 144.04a	*p* < 0.05	n.d.	n.d.	n.d.	
Vitexin	431	311	431→311	75.78 ± 4.48c	1966.23 ± 246.04b	5263.67 ± 352.42a	*p* < 0.05	15.60 ± 1.58c	303.08 ± 8.66b	1050.46 ± 63.59a	*p* < 0.05
Total flavones				87.76 ± 6.03a	2385.92 ± 269.09b	6707.32 ± 206.86a	*p* < 0.05	15.60 ± 1.58c	303.08 ± 8.66b	1050.46 ± 63.59a	*p* < 0.05
Proanthocyanidins											
Procyanidin B_2_-3-*O*-gallate	729	577, 289	729→577	65.68 ± 4.47b	67.15 ± 0.24b	139.11 ± 4.20a	*p* < 0.05	n.d.	n.d.	n.d.	
Procyanidin B_2_	577	425, 407, 289	577→425	48.03 ± 0.39c	176.24 ± 5.77b	491.72 ± 20.12a	*p* < 0.05	n.d.	n.d.	n.d.	
Total proanthocyanidins				113.71 ± 4.86c	243.39 ± 6.01b	630.83 ± 24.31a	*p* < 0.05	n.d.	n.d.	n.d.	
Total phenols compounds				1420.89 ± 3.06c	3640.28 ± 234.93b	8407.83 ± 163.85a	*p* < 0.05	54.79 ± 3.12c	317.42 ± 9.93b	1075.82 ± 62.05a	*p* < 0.05

Data are expressed µg/g dw and are means ± SD of two extraction and duplicate chromatographic analysis. Statistical analysis was by one-way ANOVA with Tukey’s post hoc test (different letters indicate significant differences). Q. T.: Quantification transition.

**Table 4 foods-12-02047-t004:** Total antioxidant capacity (TAC) and ferric reducing antioxidant power (FRAP) of unsprouted (BW-0) and sprouted buckwheat (48 h, BW-48; 72 h, BW-72).

	BW-0	BW-48	BW-72
TAC	26.21 ± 4.23b	31.09 ± 2.81b	46.99 ± 3.21a
FRAP	9.91 ± 0.45c	13.63 ± 0.20b	17.98 ± 0.40a

TAC and FRAP are expressed as µmol trolox eq/g dw. Data are means ± SD of two different extractions and triplicate spectrophotometric analysis. Statistical analysis was by one-way ANOVA (always: *p* < 0.05) with Tukey’s post hoc test (different letters indicate significant differences).

**Table 5 foods-12-02047-t005:** Metabolomic profile of unsprouted (BW-0) and sprouted buckwheat (48 h, BW-48; 72 h, BW-72).

Metabolites	ppm (δ)	BW-0	BW-48	BW-72
Tryptophan	7.735 (d), 7.754 (d), 7.265 (t), 7.201 (t)	90 ± 1.25b	106 ± 12.7b	130 ± 12.7a
Phenylalanine	7.431 (m), 7.379 (m), 7.344 (d)	719 ± 53a	772 ± 50a	737 ± 26a
Tyrosine	7.194 (d)	269 ± 51.4a	280 ± 10a	272 ± 20a
Glutamine	2.475 (m), 2.155 (m)	1305 ± 32b	1870 ± 211b	3876 ± 159a
Alanine	1.480 (d)	570 ± 14b	770 ± 12a	719 ± 50a
Valine	1.042 (d), 0.978 (d)	213 ± 21b	384 ± 38a	344 ± 68a
Isoleucine	1.010 (d), 0.950 (t)	183 ± 11b	284 ± 36a	237 ± 63a
Sucrose	5.412 (d)	872 ± 202a	1451 ± 796a	1536 ± 771a
Glucose	5.237 (d)	300 ± 20.2b	2476 ± 202a	2320 ± 202a
Fructose	4.107 (d)	753 ± 397b	1044 ± 329ab	1581 ± 605a
Acetate	1.949 (s)	291 ± 32b	310 ± 32ab	330 ± 27a
Lactate	1.331 (d)	329 ± 5b	519 ± 55a	497 ± 79a
GABA	3.023 (t), 2.313 (t), 1.905 (m)	987 ± 17a	985 ± 29a	955 ± 30.6a
NADP^+^	9.341 (s), 9.103 (d), 8.841 (d), 8.452 (s)	24.5 ± 2.06a	20.8 ± 2.65a	20.4 ± 3.26a
Trigonelline	9.090 (s), 8.840 (m)	68 ± 4a	52 ± 11ab	47 ± 13b

Data are expressed as signal area and are means ± SD of five spectroscopic analyses. Statistical analysis was by one-way ANOVA (tryptophan, glutamine, alanine, valine, isoleucine, glucose, fructose, acetate, lactate, trigonelline: *p* < 0.05) with Tukey’s post hoc test (different letters indicate significant differences). For convenience, only one signal is reported for sucrose, glucose, and fructose. The letters in brackets indicate multiplicity (s: singlet; d: doublet; t: triplet; m: multiplet). GABA: γ-aminobutyric acid; NADP^+^: nicotinamide adenine dinucleotide phosphate.

**Table 6 foods-12-02047-t006:** Phytic acid content of unsprouted (BW-0) and sprouted buckwheat (48 h, BW-48; 72 h, BW-72).

	BW-0	BW-48	BW-72
Phytic acid	13.34 ± 0.27a	12.09 ± 0.38ab	11.38 ± 0.34b

Data are expressed as mg/g dw. Data are means ± SD of two different extractions and triplicate spectrophotometric analysis. Statistical analysis was by one-way ANOVA (*p* < 0.05) with Tukey’s post hoc test (different letters indicate significant differences).

**Table 7 foods-12-02047-t007:** Pepsin, trypsin, and chymotrypsin activity in the absence or presence of unsprouted (BW-0) and sprouted buckwheat (48 h, BW-48; 72 h, BW-72).

	w/o BW Extract	BW-0	BW-48	BW-72
Pepsin activity	89.44 ± 4.81b	92.41 ± 5.16b	107.41 ± 20.50ab	152.22 ± 27.76a
Trypsin activity	180.37 ± 49.83a	6.39 ± 2.03c	65.56 ± 14.35bc	82.96 ± 8.26b
Chymotrypsin activity	540.28 ± 71.10a	360.54 ± 62.22a	372.78 ± 63.79a	359.81 ± 46.65a

Pepsin, trypsin, and chymotrypsin activities are expressed as U/mg enzymes. Data are means ± SD of two different extractions and at least triplicate activity measurements. Statistical analysis was by one-way ANOVA (pepsin and trypsin: *p* < 0.05) with Tukey’s post hoc test (different letters indicate significant differences).

## Data Availability

No new data were created or analyzed in this study. Data sharing is not applicable to this article.

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
