# Peer review of "Effect of Sprouting on Biomolecular and Antioxidant Features of Common Buckwheat (Fagopyrum esculentum)"

_foods, 2023, doi:10.3390/foods12102047_

Round 1

Reviewer 1 Report

This manuscript studied the effect of sprouting on biomolecular and antioxidant features of common buckwheat. The research is interesting and plenty of data are presented in the manuscript. however, there are some questions that should be addressed in the list below.

Introduction

Various study about effect of sprouting or germination on nutritional and healthy properties of common buckwheat has been reported, thus author should emphasize the novelty of this study in introduction section.

Materials and Methods

Line 101: A space is missing between “amino” and “acids”.

Results and Discussion

In Fig 2A & C, each lane represents what (BW-0, BW-48, BW-72) should be labeled.

Line 218-229: For the discussion of the results, the authors made assumption based on other publications. Is it possible to have in vitro starch digestibility testing experiments for this assumption?

Line 293: When the abbreviations (SFA, MUFA) firstly appear, the whole name should be listed. Please check the whole manuscript.

Line 302-307: It is suggested to move the notes to the end of Tables. Please check the other Tables in the manuscript.

Line 310-311: If possible, it is better to present the data of the concentration of CD using table (for instance adding a line in Table 1) or figure.

Similar issue in line 317-319, it is suggested to put the data of phytic acid content in Table 2.

It is suggested to move the section of “3.4. Anti-nutritional factors” to the end of “Results and Discussion” part.

Line 391-397: Because there has already been lots of research on the increase in polyphenols during germination, the authors should better and deeper discuss it more.

Line 398-405: It is suggested to move this paragraph to the section of “3.6. Antioxidant capacity”.

Line 406-412: The discussion of “3.6 Antioxidant capacity” should be improved. Similar issue in section of “3.7. Metabolome”.

Conclusions

Only two germination conditions (48 and 72 h) were studied in this research, it is hard to draw the conclusion of “72 hours appeared the best sprouting duration.”

no

Reviewer 2 Report

Comments

- The purpose of the study is not sufficient and clear. should be developed

-There are lot of typing errors, and should be corrected

-More information about materials should be given..

-In material section: more detailed information about buckwheat material should be given. Maturation? Location? Harvest time?

-Text should be corrected according to Instruction..

-The introduction should be developed and given in appropriate paragraphs according to the subject flow.

-The aim of study should be simpler

-Why was aqueous extraction applied? Solvent and/or mixture in which phenolic compounds are most soluble, methanol:water mixture?

-What was the final moisture content of the material dried at 50 oC for 8 hours? Were the results calculated according to these moisture values?

-How was oil obtained from 0.1 g of buckwheat powder? Isn't this amount too small?

-Why was only FRAP preferred as antioxidant activity?

-Many reference should be added and discussed with results.

Good in general. Ma text should be checked thoroughly for some typing errors.

Reviewer 3 Report

The manuscript “Effect of sprouting on biomolecular and antioxidant features of common buckwheat (Fagopyrum esculentum)” describing the presence of several bioactive molecules in sprouting buckwheat. The study is valuable, however, requires some improvements.

1. Introduction

This part can be extended since is quite short and contain only 8 references, while many more different types of molecules were determined in this study.

2. Methods

- “2.9. Lipid content and composition” – “Total lipids were extracted from 0.1 g of buckwheat powder” - the lipid content determination in 0.1g of the sample which contains a very low content of oil is burdened with a large measurement error. For e.g. for phenolic compounds was used 2 g. Can you explain it?

- please provide a source of FAME;

- it is not clear how the lipids were extracted for e.g. used solvent;

- please provide a source of α-tocopherol;

- if only α-tocopherol was used as a standard I just wondering how the other tocopherols were identified?

3. Results and discussion

“3.5.1. Tocols”

- “Buckwheat exhibits levels of tocopherols similar to wheat, barley, oats, and rye, with γ-tocopherol being the main isoform present, typically in a > 10-fold excess with respect to α- and δ- tocopherols [54].” - I disagree with this statement. In my opinion, the reader does not get the most important information, namely that wheat, barley, oats, and rye contain tocotrienols while buckwheat does not. Please see for instance:

“Lipophilic bioactive compounds in the oils recovered from cereal by-products. https://doi.org/10.1002/jsfa.7511”.

- “tocopherols isomers” - exist only two isomers of tocopherols (beta and gamma), therefore please instead of "isomers" use "homologues". For more info about tocopherols please see for e.g. “Free and Esterified Tocopherols, Tocotrienols and Other Extractable and Non-Extractable Tocochromanol-Related Molecules: Compendium of Knowledge, Future Perspectives and Recommendations for Chromatographic Techniques, Tools, and Approaches Used for Tocochromanol Determination. 

- Below Table 3, please provide how each homolog of tocopherols was quantified, since I understood from the method part it was used only alpha-tocopherol for creating a calibration curve. Yes? Why this is so important? The slope for homologue alpha is twice and more times lower than for other homologues (please see for e.g. positions listed below). This means that if other homologues are quantified based on alpha-tocopherol, they are overestimated by about two times.

“Rapid baseline-separation of all eight tocopherols and tocotrienols by reversed-phase liquid-chromatography with a solid-core pentafluorophenyl column and their sensitive quantification in plasma and liver. https://doi.org/10.1016/j.chroma.2012.04.042”;

“An alternative RP-HPLC method for the separation and determination of tocopherol and tocotrienol homologues as butter authenticity markers: A comparative study between two European countries. 

4. Conclusions

This part is well written.

s
